# communications
# engineering

# Deposition of ultra-thin coatings by a nature-inspired Spray-on-Screen technology

Rachith Shanivarasanthe Nithayananda Kumar [1,2 ✉], Andrea Valencia Ramirez [1,2], Pieter Verding [1,2], Philippe Nivelle[1,2], Frank Renner [1,2], Jan D'Haen [1,2] & Wim Deferme [1,2 ✉]

Nanometre-thick, ultrathin coatings applied over a large area are of paramount importance for various application fields such as biomedicine, space and automotive, organic electronics, memory devices, or energy storage devices. So far wet chemical deposition as a cost-effective, scalable, and versatile method can only be used for thicker deposits. Here the formation of uniform ultra-thin coatings with thicknesses below 15 nm using a nature-inspired, roll-to-roll compatible Spray-on-Screen (SoS) technology is reported. For this, the finite micro-droplet generation of Ultrasonic Spray Coating (USSC) is combined with the coating formation from a screen printing mesh. Hydrophobic micro-threads of the mesh, resembling the micro-hair on the legs of water striders, produce millidroplets from micro droplets, and when applying an external pressure to the mesh, dynamic wetting is enforced. The proposed technology is applicable for a wide variety of substrates and applications. It is shown by theory and experiment that ultra-thin coatings below 5 nm homogeneous over a large area can be deposited without the use of extended ink formulation or high substrate temperatures during or after deposition. This simple yet effective technique enables the deposition of ultra-thin films on any substrates, and is very promising to fabricate the organic, inorganic electronics devices and batteries cost effectively.

[1] Institute for Materials Research (IMO-IMOMEC), Hasselt University, 3590 Diepenbeek, Belgium. [2] IMEC vzw, Division IMOMEC, Wetenschapspark 1, B-3590 Diepenbeek, Belgium. ✉email: rachith.shanivarasanthe@uhasselt.be; wim.deferme@uhasselt.be

Growing research on ultra-thin coatings is essential due to its diverse field of applications which includes bio-compatible coatings[1] in biomedical devices, ultra-low friction and wear-resistant thin films in space and automotive fields[2], functional thin coatings in thermoelectric[3], photovoltaics[4], light-emitting diodes[5], sensors[6], spintronics[7], lasers[8], batteries[9], super capacitors[10] etc. Conventionally, fabrication of these thin coatings includes expensive and time-consuming vacuum deposition techniques like sputtering[11], thermal[12] and e-beam evaporation[13], atomic layer deposition[14], molecular beam epitaxy[15], metal-organic chemical vapour deposition[16] etc. Solu-tion processable techniques like spin coating[17] offer a unique solution of achieving uniform thin coatings[18] at a low cost. Spin coating however has one major disadvantage: it lacks the cap-ability of upscaling to big areas. Other solution processing tech-niques like dip-coating[19], inkjet printing[20,21], slot-die coating[22], doctor blade coating[23], screen printing[24], spray coating[25], spray pyrolysis[26] are roll-to-roll (R2R) compatible but they are incap-able of depositing ultra-thin coatings (<15 nm) without extended ink formulation or heat treatments[20], which is not suitable for temperature-sensitive substrates. In Supplementary Fig. 1, dif-ferent solution processing techniques are depicted showing the relationship between scalability and minimal layer thickness as described in the literature. Droplet-based technologies such as inkjet printing and spray coating need an optimized ink for-mulation and a thermal heat treatment to achieve ultra-thin coatings and have the problem of coffee rings and Marangoni flow[27]. Depositing droplets on surfaces leads to a inhomogeneous surface coverage by the coffee ring[28] effect. If the droplets are larger than the capillary length, then gravity dominates the wet-ting behaviour and might deform the soft substrates. If not, (microdroplets), then the surface tension of the liquid dominates which would hinders the nature of wetting. However, the latter could be overcome by forced dynamic wetting, which could allow partial wetting of the liquid to spread uniformly over the substrate and this occurs when the liquid velocity is set by an external force, unlike the equilibrium wetting induced by the capillary force[29]. Since the microdroplets evaporate very fast (~30 ms)[30,31], by the time an external force is being applied to initiate the dynamic wetting, the droplets would evaporate and again form coffee rings. Therefore, instead of depositing a microdroplet directly on the substrate, if in case a thin film of liquid is deposited, then by dynamic wetting, the influence of the coffee rings and hence the irregular deposition could be eliminated. Therefore, we establish that to achieve a uniform thin film we should have a simultaneous effect of generating a uniform thin film over the substrate, and to apply the external force to activate dynamic wetting and this brings about an additional challenge.

Also here, nature has its example: water striders are water-walking insects thanks to their non-wetting legs[32–34]. These legs are covered with a large number of tiny hair[35] as shown in Supplementary Note 2 and Supplementary Fig. 2. Each of these tiny hairs have nano-grooves to possess water resistance. Due to this unique combination of hierarchical micro and nanostructure on the legs surface, the dimples created by the legs on the water can be up to 4.38 ± 0.02 mm in depth, without piercing the water, which is significantly greater than the capillary length of water ~2.7 mm[35]. Therefore, a combination of hierarchical microstructure could shear the liquid without piercing. This ingenious technology in nature has inspired us to use non-wetting microstructures for a dual purpose: (1) convert the microdroplets into milli-droplets (microdroplet coalescence) and (2) induce dynamic wetting of the liquid, without breaking the surface of the liquid. Hence, a uniform spreading of the thin liquid film over the substrate thereby results in the formation of a uniform ultra-thin coating of the selected material over any given substrate.

After a theoretical introduction on the Spray-on-Screen tech-nology (SoS), this article will show the deposition of ultrathin coatings with the SoS technology measuring the thickness and roughness of the final coatings. For a water based poly(3,4-ethylenedioxythiophene) polystyrene sulfonate (PEDOT:PSS) formulation, isoproponal alcohol (IPA) based polyethylenimine, 80% ethoxylated (PEIE) ink and IPA based Zinc oxide (ZnO) nanoparticle formulation, we achieved the thickness of 15.8 nm, 4.8 nm and 15.3 nm, respectively, with surface roughness (RMS) of 1.1 nm, 0.3 nm and 2.6 nm respectively. The unique combi-nation of droplet generation with Ultrasonic spray coating (USSC), the microdroplet coalescence and the dynamic wetting applying a screen printing mesh, mimicking nature, results in ultra-thin (<15 nm), homogeneous coatings for large area deposition, to our knowledge never shown before in the scientific literature.

## Results

**Spray-on-Screen coating technology**. To realize the idea of using nature-inspired forced dynamic wetting of the microdroplets on any substrate, we have used USSC for the generation of micro-droplets (Fig. 1b), and a screen mesh to convert the microdroplets to milli-droplets (due to its microstructure) and to facilitate the forced dynamic wetting of the deposited liquid (Fig. 1a, d). We stipulate that this thin film could best be formed above the

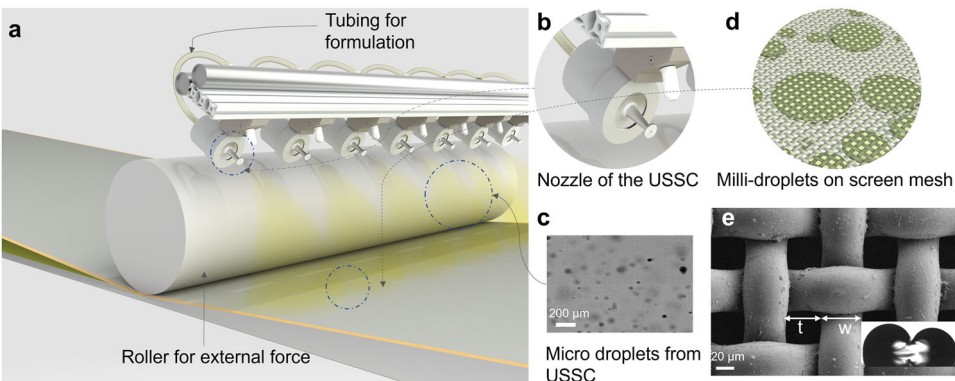

**Fig. 1 Spray on screen technology and its component. a** Graphical abstract of the complete spray on screen technology (**b**) Schematic of the nozzle used in the Ultrasonic spray coating(USSC), **c** High-speed camera image of droplets from USSC, with a diameter range of 15 μm to 90 μm. Video of USSC-generated droplets is shown in Supplementary Video 1[50], **d** Graphical illustration of conversion of microdroplets into milli droplets on the screen mesh, **e** Scanning electron microscopy image for the screen used in the present study, with t = thickness of the thread (~50 μm) and w = opening of the mesh (~40 μm), inset, shows the hydrophobic thread cutting the water without piercing the same, and resembles the tiny legs of water striders.

substrate (thus on the screen mesh) to prevent the evaporation and thus coffee ring formation on the substrate itself.

The selection of USSC is mainly due to the capability of forming homogeneous droplets less than 100 μm in diameter (Fig. 1c, and Supplementary Video 1). Further, the velocity of the generated droplets can be easily controlled by the nitrogen pressure, reducing the singularity which exists for other spray coating technologies; i.e., velocity and/or temperature-induced droplet interaction with the mesh of the screen[36]. Details of the working of USSC is given in Supplementary Note 1. A polyester mesh, as can be seen from the SEM image in Fig. 1e, is typically applied for screen printing, with a dimension of the thickness of the wire (t) ~50μm and the interspacing between the wires (w) ~40μm was used for a dual purpose: to convert the microdroplets into milli-droplets and to facilitate the forced dynamic wetting of the deposited liquid as is explained in more detail before. Inset of Fig. 1e shows the hydrophobic nature of the polyester mesh, which mimics the water striders leg, where it allows for the forced dynamic wetting without piercing the liquid of interest. The polyester screen mesh was cleaned thoroughly (details of cleaning is given in materials and methods) before the conduction of the experiments and possessed an equilibrium water contact angle of 104.6º (Supplementary Fig. 3) details of the contact angle measurements is given in Supplementary Note 3. The present mesh could be considered as a substrate with textures. As explained by Marmur et al.[37] the wetting on a rough surface (or in our case textured substrate) assumes multiple local free energy minima with different contact angles. Finally, the overall free energy minima could follow the traditional contact angle theory values once the droplet is much larger than the roughness scale of the substrate underneath. Also, since the periodic textures on the surface involve multiple pinning points, there would be multiple local minima of free energy and hence the obtained water contact angle is different from the substrate made up of polyester without any texture. However, this analysis would be entirely different for the droplet whose dimensions are less than the capillary length and as in our case equal or smaller than the throat (w) of the mesh. In here, the four corners of the mesh act as individual pinning points and act as local minima of free energy. Also, in our case, the pressure exerted by the nitrogen gas upon the droplets makes the analysis of free energy a complex phenomenon. Therefore, understanding of the global free energy minima to attain the equilibrium is a complicated state of affair. In the following section, we explain the process of reaching the global minima on the mesh and the process of forced dynamic wetting by the mesh.

### Theoretical and experimental approach for the technology
*Theoretical approach for the ultra-thin film deposition.* As thousands of droplets are being generated by USSC hitting the mesh within the same time frame, these micro-droplets coalesce before they evaporate and form milli-droplets (Supplementary Note 4 and Supplementary Fig. 4). The formation and profile of the milli-droplets upon coalescence of microdroplets depends on the spreading coefficient and the geometry of the surface texture. The spreading coefficient (S) is usually calculated to distinguish between different wetting states (Eq. 1). In general, it represents the surface energy difference between partial and complete wetting stages (Fig. 2g), i.e., at partial wetting, θ would be in between $90^0$ and $180^0$ and at complete wetting, θ would be $180^0$. Therefore:

$$S = \gamma_{sv} - (\gamma_{sl} + \gamma_{lv}) \qquad (1)$$

Where, $\gamma_{sl}$, $\gamma_{sv}$ and $\gamma_{lv}$ are the surface tension at solid/liquid, solid/vapour and liquid/vapour region respectively. If the spreading coefficient is equal to or more than zero, then the droplet completely wets the surface and if it is less than zero then the droplet partially wets the surface. Based on the surface tension of the formulation and the surface energy of the mesh material, S can be determined. In the present work, a low surface energy polyester mesh (43 mN m$^{-1}$)[38] and two different formulations at opposite ends of the surface tension spectrum were studied.

For water as a solvent ($\gamma = 72$ mN m$^{-1}$), being a high surface tension liquid, water would inevitably result in partial wetting. Therefore, there could be two situations, a) the deposited microdroplets would coalesce into a sizable hemispherical milli-droplet, b) the microdroplets would be pushed towards the downside of the mesh by the applied nitrogen pressure. The latter would happen only if the diameter of the droplet is smaller than the throat area of the mesh. The former resembles the drop sitting on top of air pockets and the growth and profile of the droplet are dominated by the roughness of the surface texture. In this case, (Fig. 2d, inset top right of Fig. 2e), where a drop is sitting on top of the pockets, the contact angle relation is given by the Cassie–Baxter model (Fig. 2d)[39–41] $cos(\theta^*) = r\ \varphi_s\ cos(\theta_E) - (1 - \varphi_s)$, where $(1 - \varphi_s)$ is the area made by the liquid-air fraction. As $\varphi_s \geq 0$ the apparent contact angle $\theta^* \geq 180°$ even for a surface whose $\theta_E < 90°$.

On the other hand, for a low surface tension liquid like isopropanol alcohol (22 mN m$^{-1}$), wetting would be complete; i.e., it will completely wet the threads of the mesh. Any further deposition would result in increasing the thickness of the deposited droplet on the mesh's thread, which grows until the droplet from two neighbouring mesh threads reach the critical thickness (w/2, w is the width of the throat of the mesh) wherein coalescence would result in a continuous liquid film between the mesh. Upon further deposition of the microdroplets, the continuous thin film between the mesh would keep spreading over the neighbour thread of the mesh and simultaneously grow both in width and thickness. The milli-droplet would acquire the hemispherical shape, with the top half being above the mesh and the bottom half being below the mesh (droop $R_d$), as is shown in (Fig. 2b, inset top right). The curvature of the droplet is governed by the Laplace equation and the curvature is the same on the top and bottom of the mesh[36,42]. The Laplace equation relates the pressure inside the droplet to its curvature. Further increase in droplet deposition increases the width and thickness of the coagulated droplet until it reaches the capillary length. For the fully wetted state, like for the case of IPA wetting on the mesh (liquid of low surface tension), (Fig. 2a), the relationship between the apparent or macroscopic contact angle ($\theta^*$) and the equilibrium contact angle ($\theta_E$) of a smooth flat substrate is given by the Wenzel equation[43], $cos(\theta^*) = r\ cos(\theta_E)$, where r is the roughness parameter that describes the ratio between the actual area wetted by the liquid (including the pores) to the area projected onto a flat substrate. Therefore, the spread-ability of the deposited liquid over the mesh depends on the surface tension of the solvent being used.

Since the spreading coefficient predicts the state of wetting, we tried to understand this by dropping a microliter droplet (2 μl) on the mesh and followed the wettability of the droplet over the mesh. Depending on the surface tension of the liquid, the wetting behaviour on the mesh is altered. As can be seen from Fig. 2b, and inset top right, the low surface tension liquid IPA completely wets the mesh (hydrophilic) and liquid droops down the mesh achieving the same length, thickness and curvature as the liquid on top of the mesh, (Fig. 2c and inset top right) (zoomed bottom image showing the complete wetting of thread by IPA). If a substrate is placed underneath the mesh between, $0 \geq x$, the droplet touches the substrate. Here, x is the distance between the substrate and the bottom of the droplet, $R_d$ is the droplet diameter and D is the thickness of the mesh.

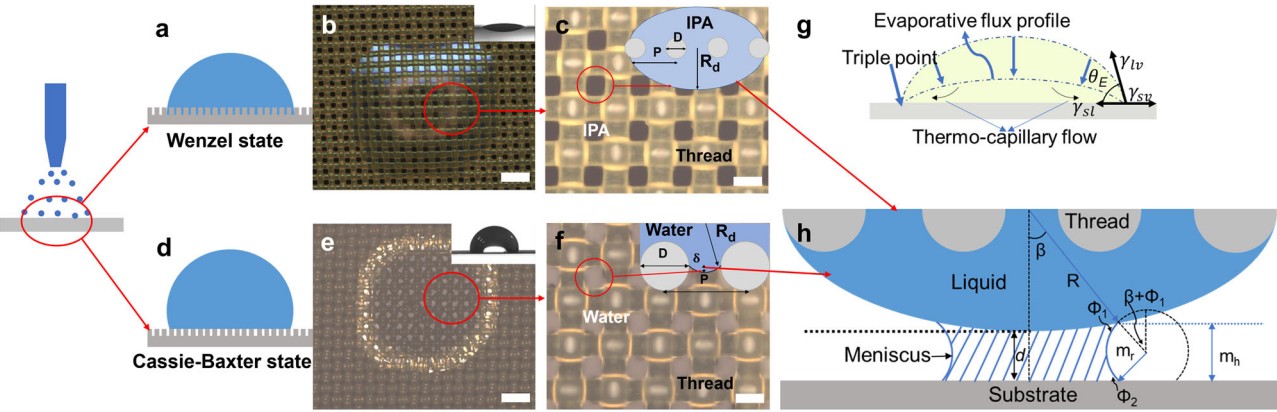

**Fig. 2 Droplet behaviour on the mesh. a** Liquid wetting an idealized rough surface with Wenzel state, **b** optical microscope image of Isopropanol alcohol (IPA) on mesh displaying behaviour of low surface tension liquid, inset shows IPA in Wenzel wetting state, **c** curved profile formation of IPA over the mesh (top as well as bottom), resulting in drooping below the mesh. **d** Liquid wetting an idealized rough surface with Cassie–Baxter state, **e** optical microscope image of Water on mesh displaying the behaviour of high surface tension liquid exhibiting Cassie–Baxter state of wetting on micro-mesh (inset top right), **f** partial drooping of water with a meniscus profile of the water between the threads of the mesh with a definite contact line. **g** Force balance at the triple junction between a liquid drop and flat substrate[51] and profile of the deposited droplet, with an equilibrium contact angle (θ), triple point, and also the evaporative flux profile[52] **h** drooped liquid with radius $R_d$ interacting with the substrate forming meniscus of radius $m_r$, the height of the meniscus is $m_h$, filling angle is β, $\Phi_1$ and $\Phi_2$ are the contact angle of the liquid meniscus with the drooped liquid and with the substrate, d is the distance between the drooped liquid and substrate. (Scale 200 μm).

With water as a solvent, as seen in Fig. 2e and inset top right, the hydrophobic behaviour is exhibited. This could be seen as a droplet sitting on air pockets, which resembles the Cassie–Baxter state of wetting, (Fig. 2d). Water forms a meniscus around the thread. At equilibrium, the maximum droop of the droplet would be at the centre of the mesh opening, with a droop length of $\delta = (\sqrt{2}\ P\text{-}D)^2/8\ R_d$[36], where, $R_d$ is the radius of the droplet, $D$ is the thickness of the mesh and $P$ is the pitch. The droop of water for a 1-millimetre diameter droplet could be seen in Fig. 2f. As the droplet diameter reduces, the droop length increases. If finally, δ *becomes* greater than *D/2*, then the droplet touches the substrate underneath the mesh, (Fig. 2f and inset top right).

Even as the solvents being used are at the opposite end of surface tension, once the droplet touches the substrate, there exist two situations that come to light once the drooped liquid on screen touches the bottom substrate: The first situation is where, the capillary force that arose between the substrate, the liquid on the mesh and the air interface would result in the spreading of the liquid onto the substrate to reach the global minimum of the substrate's surface energy. This initiates a hydrodynamic flow of the droplet[44]. Once the thin liquid film formed on the mesh touches the bottom substrate, the liquid will be drawn from the thin film and would form a meniscus between the mesh and the substrate (inset Fig. 2c, f). An attractive force is generated by the formed meniscus. The attractive force is mainly due to the existence of the surface tension of the liquid around the periphery of the meniscus and due to the reduced pressure inside the meniscus as compared to the capillary pressure. This attractive force results in the second situation where attractive force pulls the mesh towards the substrate and drains the liquid above the mesh. By assuming the height of the meniscus $m_h$ to be smaller than the capillary length $l_c$, the capillary force (F) is given by (Eq. 2)[44] and it can be seen in Fig. 2h,

$$F = 2\pi\gamma R_d\left(2c - \frac{x}{m_r}\right) \tag{2}$$

where γ is the surface tension of the liquid, $R_d$ is the droplet radius, β is the filling angle, $\Phi_1$ and $\Phi_2$ are the contact angle of the liquid meniscus with the drooped liquid and with the substrate respectively, $x$ is the distance between the drooped liquid and

substrate and constant $c$ can be calculated from the Eq. 3, which is given by:

$$c = \frac{\cos(\beta + \Phi_1) + \cos(\Phi_2)}{2} \tag{3}$$

This force created would then be applied by the screen mesh on the fluid which influences the wetting behaviour and hence forced dynamic wetting will happen. Also, as there would be many milli-droplets formed due to the microdroplets, these individual droplets would simultaneously try to wet the substrate (Fig. 1d). Finally, these multiple droplets will form a coating with an abrupt increase in the amplitude of the coating thickness when the droplets deposited on the screen interfere constructively in the substrate. Fortunately, the presence of the screen mesh prevents the formation of abrupt thin liquid coatings, due to the applied force by the screen mesh over the liquid film, which resembles the force exerted by the nano-hairs of the water striders on a liquid. The hydrostatic pressure inside the liquid increases, which forces the liquid to flow in lateral dimensions. This induces dynamic wetting, which follows the hydrodynamic flow induced by the capillary force. The combination of capillary force and the external force applied by the mesh would assure that the liquid wets the surface irrespective of the solvent and type of substrate being used.

*Experimental approach for the ultra-thin film deposition.* To corroborate the flexibility of the SoS on the formation of ultra-thin coatings, PEDOT:PSS, a water based material, 1:10 by volume, PEIE in IPA, 0.1:35 by volume, and a polymer-metal oxide composite of ZnO nanoparticles with a diameter of 10–15 nm mixed with PEIE in IPA as a solvent, 1:0.1:35 by volume, is deposited on Indium Tin Oxide substrates (ITO). ITO substrates with a size of 2.5 × 2.5 cm² in size are cleaned according to the cleaning protocol given in material and methods and UV-treated for 30 min before being deposited by the formulation of interest. The surface energy of the ITO is measured and calculated by using Fowke's theory[45], and it was found to be 117.37 mN m⁻¹. The obtained results are compared with standard spin-coated layers of the same materials on ITO substrates.

**Table 1 Deposition parameters for the various materials and the solvents used.**

| | Substrate temp °C | Solution concertation | Number of passes | Nitrogen pressure | Flow rate ml min⁻¹ | Path speed mm s⁻¹ |
|---|---|---|---|---|---|---|
| PEDOT:PSS | 60 | 1 ml PEDOT:PSS/15 ml water | 1 | 1.5 kPa | 0.5 | 10 |
| PEIE | 30 | 0.01 ml in 30 ml IPA | 1 | 1.5 kPa | 0.8 | 10 |
| ZnO+PEIE | 30 | 2 ml ZnO+0.01 ml PEIE + 50 ml IPA | 1 | 1.5 kPa | 0.9 | 10 |

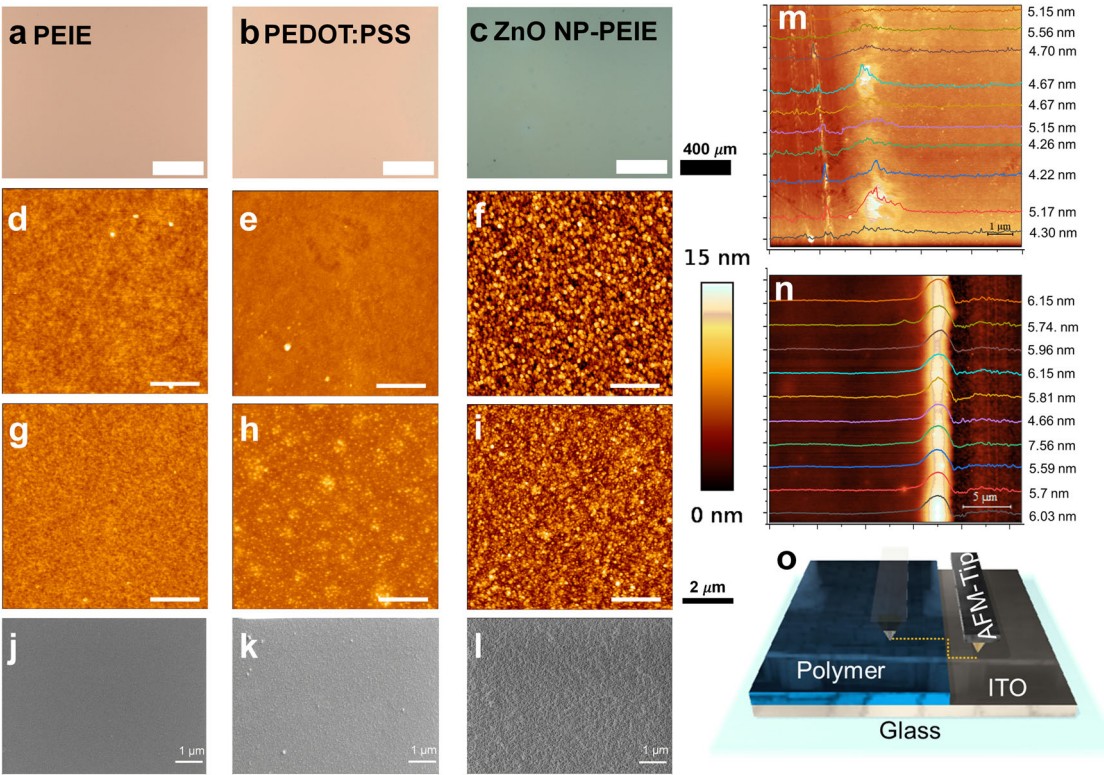

**Fig. 3 Thin film forming capability of the spray on screen technology. a–c** Optical microscopy image, **d–f** Atomic force microscopy for the PEDOT:PSS, PEIE and ZnO-PEIE composite deposited on ITO substrate. Atomic force microscopy (**g**, **h**) and (**i**), relative value for the colour scale of all the AFM images are given from 0–15 nm, and scanning electron microscopy (**j**, **k**) and (**l**) for the thin films of PEDOT:PSS, PEIE and ZnO-PEIE composite over the super yellow substrate. The sequence of the thickness measured for the spray on screen deposited PEIE on small and large samples are shown in (**m**) and (**n**), respectively. **o** graphical representation of the scratch measurement using AFM tip.

The deposition conditions for all three cases are given in Table 1. The distance between the substrate and the nozzle of the USSC and the room temperature is kept constant for all sets of experiments at 6 cm and 22 °C, respectively. And the distance between the screen mesh and the substrate is kept above the droop length of both the solvents, i.e., at 140 μm. Upon the deposition of droplets on the screen, the screen is made to touch the substrate so that the capillary force would draw the liquid from the top of the screen and would induce forced dynamic wetting. It should be noted that the mesh, which is held by capillary force, should be released from the substrate otherwise the deposited material would be deposited only at the region where the mesh thread overpassing takes place (Supplementary Note 5 and Supplementary Fig. 5).

To begin with, the uniformity of the deposited layers over the large area is investigated by optical microscopy and as is clearly seen from Fig. 3a–c, irrespective of the materials and solvents being used, the deposited layers show a uniform deposition without any coffee rings. Because of the absence of irregularities in the thickness of the deposited coating over a large area, this technique promises a new regime of upscalable fabrication

technology for ultra-thin films, provided the obtained thickness is less than 15 nm.

To determine the thickness of the different materials, tapping mode AFM is applied[46] and the sequence of measuring the thickness by AFM is given in Fig. 3o and in Supplementary Figs. 6–8, details of the thickness measurement is given in Supplementary Notes 6–8. As an example, in Fig. 3m, n, the measured thickness which is calculated by utilizing the profile over the scratch 10 times is shown for PEIE deposited on small (2.5 × 2.5 cm²) and large substrates (5 × 5 cm²) respectively (Supplementary Notes 9 and 10 and Supplementary Figs. 9 and 10). To measure the thickness, all materials are deposited on ITO coated glass slides and the concentration of the solution is varied by keeping all other parameters constant. The desired ultra-low thickness of the material of interest was achieved by varying the solution concentration and the flow rate of the solution. PEDOT:PSS being diluted in water evaporated less during the flight from the nozzle to the screen and hence the flow rate was less compared to IPA based materials (Table 1). Also to evaporate the deposited thin liquid film for PEDOT:PSS, the substrate temperature was kept at 60 °C, which is also more than the

**Table 2 Minimum thickness achieved for the different materials over the ITO substrate.**

|  | PEDOT:PSS | PEIE | ZnO + PEIE |
|---|---|---|---|
| Minimum thickness | 15.8 ± 0.6 nm | 4.8 ± 0.1 nm | 15.3 ± 0.6 nm |
| Surface roughness (RMS) | 1.1 nm | 0.36 nm | 2.6 nm |

substrate temperature used for IPA based materials, which was 30 °C. Excess of solution flow rate resulted in an edge effect[47,48], where more material was accumulated. Therefore, the flow rate was optimized according to the evaporative properties of the solvent being used. The minimum thickness achieved for PEDOT:PSS, PEIE and ZnO-PEIE is 15.8 ± 0.6 nm, 4.8 ± 0.1 nm and 15.3 ± 0.6 nm, respectively, as is shown in Table 2, the elemental analysis for the PEDOT:PSS and PEIE-ZnO nanocomposite on the ITO coated glass substrate is shown in Supplementary Figs. 11 and 12, details of the elemental analysis by EDX is given in Supplementary Notes11 and 12. To further investigate the effect of solution concentration on the thickness of the deposited coatings, a comprehensive investigation was done for all three materials with varying concentrations and the results are shown in Supplementary Table 1. The obtained results show that by varying the concentration of the solvent, the thickness of the deposited layer could be varied. The surface roughness (RMS) of the PEDOT:PSS, PEIE and ZnO-PEIE composite thin coating deposited on the ITO substrate is found to be around 1.1 nm, 0.3 nm and 2.6 nm, respectively, which can be seen in Fig. 3d–f. These obtained surface roughness's are comparable with the spin coated devices (Supplementary Notes 13–16 and Supplementary Figs. 13–15). This clearly shows that the Spray-on-Screen deposition has the same surface morphological properties as that of spin coated devices. Therefore, by achieving ultra-low thickness and very low surface roughness of the deposited film, this technology has filled the gap which was exciting in the up scalability of ultra-thin coating fabrication, potentially exhibiting a paradigm shift in the ultra-thin film market. Supplementary Fig. 16 shows the surface roughness comparison of the spray on screen deposited PEDOT:PSS and PEIE-ZnO nanocomposite in comparison with the bare ITO substrate. The results clearly indicate the distinction between all the substrates, the ITO being rough and upon depositing PEDOT:PSS and PEIE-ZnO nanocomposite the RMS roughness reduces. These results show that indeed we have the layer deposited on the ITO substrate.

However, the previous results have been demonstrated on a high surface energy substrate like ITO, the SoS does not render itself as the ultimate fabrication technology unless it shows its versatility on a low surface energy substrate. Therefore, to investigate the flexibility of SoS, low surface energy, temperature sensitive super yellow (SY) light emitting PPV copolymer is used as a substrate. The surface energy of the super yellow is measured by using Fowke's theory, which is around 56 mN m$^{-1}$. The details of measurements are given in Supplementary Note 17 and the results are shown in Supplementary Fig. 17. The materials and the deposition conditions used in the previous studies on ITO are being duplicated on the SY substrates. AFM measurements were performed to measure the surface morphology of the deposited PEDOT:PSS, PEIE and ZnO-PEIE composite. The obtained results are shown in (Fig. 3g–i). The surface roughness (RMS) of the deposited films was 1.1 nm, 0.8 nm and 2.6 nm, respectively, for PEDOT:PSS, PEIE and ZnO-PEIE composite, which is almost similar to the results obtained for the films deposited on the ITO substrate. SEM images obtained on the same set of samples also confirm that there are no coffee rings, and the film is uniform

throughout (Fig. 3j–l), even though the super yellow substrate was of low surface energy and soft (15.54 MPa, Supplementary Notes 18–20 and Supplementary Fig. 20). This extraordinary result confirms the independence of the Spray-on-Screen technology on the nature of the substrate. Finally, we demonstrate that this innovative Spray-on-Screen technology is perfectly suited for upscaling. Therefore, a larger 5 × 5 cm² sample was coated with PEIE, the thickness and the surface roughness of the film is measured. A thickness of 5.9 nm (Fig. 3n) and surface roughness (RMS) of 0.9 nm was achieved, and these obtained values are the same as for the small devices. The movie for the SoS is attached in Supplementary Video S2. To clearly visualize the conversion of microdroplets into milli droplets on the screen mesh, a darker formulation of PEDOT:PSS is shown in Supplementary Video S2. As can be seen clearly from the movie, the proposed Spray-on-Screen works well for larger samples as well. This demonstrates the upscale-ability of the Spray-on-Screen technology over large samples.

To display the suitability of the SoS to deposit ultrathin films on large areas, we demonstrate the deposition of an organic light emitting diode (OLED) as a potential application case, amongst many other that can be achieved with this innovative SoS technology. The methods of fabricating and characterizing the OLEDs are given in Supplementary Note 21. Since the OLED performance is highly dependent on the thickness of the various layers used, the ultrathin layer herein used is the electron injection layer (EIL). For the present case, the state-of-the-art standard device consisting of calcium as EIL deposited by thermal evaporation and also ZnO-PEIE nanocomposite as an EIL built by spin coating is compared with the SoS coated ZnO-PEIE nanocomposite. The thickness of this layer is well below 30 nm and therefore only vacuum deposition and spin coating are possible deposition techniques, apart from our SoS technology. In view of upscaling and ease of production, spin coating (only for small scale) and vacuum deposition (in need of high vacuum systems) are replaced here by the innovative Spray-on-Screen technology, capable of depositing the large area, ultrathin coatings with the ease of production of wet chemical solution processing. The structure of the solution processed materials is shown in Fig. 4a with their chemical structure in Fig. 4b–d, respectively. A slight variation in the thickness of the EIL would result in adversely affecting the OLED performance and the non-homogeneous layers, pinholes would manifest as dark spots in the illuminated devices and also, the thicker HTL or EIL would influence the electroluminescence (EL) curve due to the cavity effect. As visible from Fig. 4e, the max EL peak for all the devices is ~554 nm, this confirms the thickness of the SoS deposited PEDOT:PSS and ZnO+PEIE nanocomposite is the same as the spin coated ones. The top inset of Fig. 4e shows the illuminated small area OLEDs with SoS deposited HTL and EIL. The presence of microparticles, i.e., impurities, would create a short pathway for the charges to flow and induce dark spots, however as seen from the bottom inset of Fig. 4e, the illumination is constant throughout. The OLED performance parameters like J-V-L and EQE-V-Luminous efficacy characteristics of the device built by SoS HTL and EIL exhibit the maximum illuminance of ~15.000 lux (Fig. 4f) and a maximum luminous efficacy (Fig. 4g) of ~24lumen/watt and max EQE of 5.6% (Fig. 4g), showing that the obtained results are comparable to spin coated HTL and EIL and superior to evaporated calcium(~20 lumen/watt)[49].

Buoyed by the superior OLEDs built by SoS, large area (5 × 5 cm² substrate) OLEDs are built. Since the ultrathin layer used in the OLEDs architecture is EIL, and should be deposited over the heat sensitive low surface energy layer SY, in here the EIL was deposited by the SoS method and other organic layers are built by spin coating. The J-V behaviour of these large area

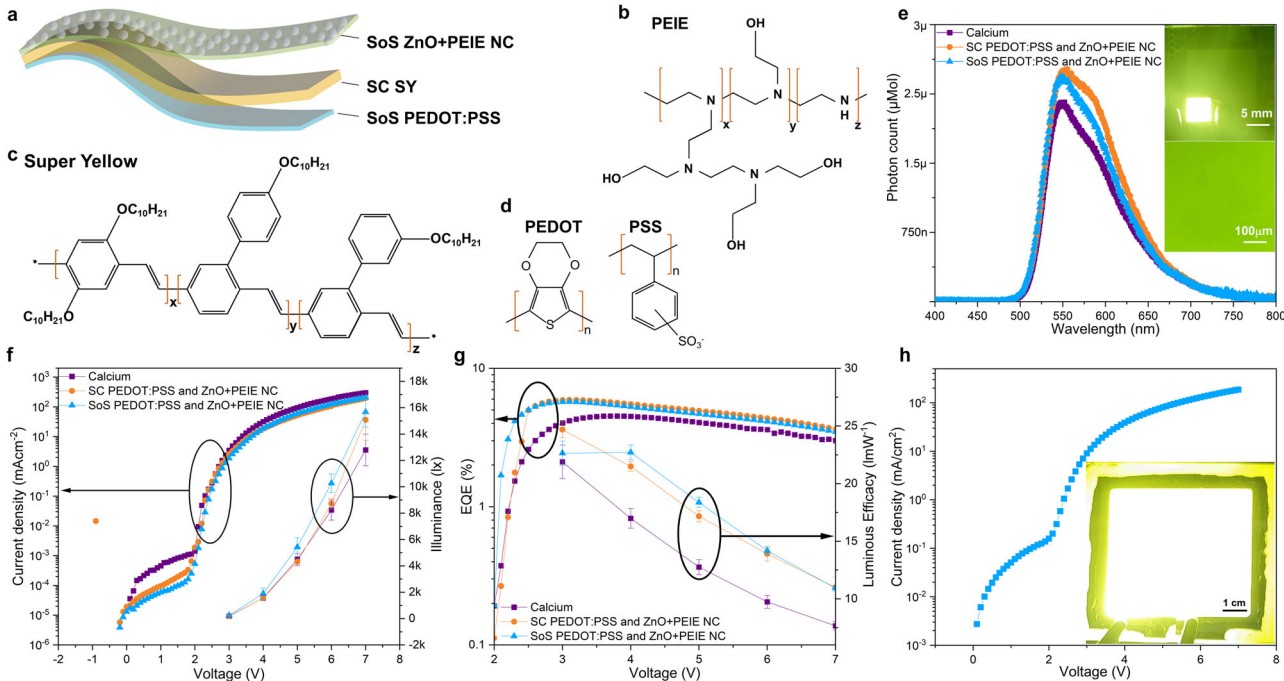

**Fig. 4 Performance comparison of Organic light emitting diodes (OLEDs) with evaporated calcium, spin coated and spray on screen(SoS) printed ZnO +PEIE nanocomposite (NC) as electron injection layer(EIL). a** Solution processed organic semiconducting materials, PEDOT:PSS as hole transport layer(HTL), Super yellow(SY) as the active layer and ZnO+PEIE NC as EIL. **b**–**d** Chemical structure of the PEIE, SY and PEDOT:PSS respectively, where X represents the polymer backbone, Y and Z represents substituents attached to the main polymer backbone. **e** Electroluminescence behaviour of EIL with various deposition methods, the top inset exhibits the illuminated small area OLEDs ($5 \times 5$ mm$^2$), where HTL and EIL and is deposited by SoS technology, bottom inset: microscopic image illustrating constant illumination. **f** Current density-voltage and illuminance (J-V-L) characteristics, **g** external quantum efficiency (EQE)-voltage and luminous efficacy characteristics of the small area OLEDs. **h** J-V characteristics for the large area OLEDs ($5 \times 5$ cm$^2$ substrate), where the EIL is fabricated using the SoS method, bottom inset: homogeneous illumination of large-area OLEDs without pinholes.

devices (Fig. 4h) is as good as small area devices (Fig. 4f) showing the uniformity in the charge injection behaviour. The inset of Fig. 4h clearly shows the homogenous illumination in the large area device, where Supplementary Video 3 shows the illumination behaviour of the small and large area devices over various applied potentials. The culmination of the theoretical and experimental explanation to produce an ultrathin film of various materials with a highly efficient large area application of OLEDs indicates the credentials of the SoS technology to be used in many large area ultrathin layer applications.

## Conclusion

The Spray-on-Screen technology inspired by the eyelids and the legs of water striders made it possible to convert the micro-droplets from USSC into milli-droplets on the screen mesh. Based on the spectrum of the surface tension of the liquid being used, the wetting of the screen is explained by the models given by Wenzel and Cassie–Baxter. This is followed by transferring the milli-droplets from the screen mesh to the substrate upon the application of an external force on the mesh. A capillary force exists between the substrate and the milli-droplets facilitating the spreading of the liquid over the substrate with the assistance of the force applied by the mesh. The presence of the screen mesh induces dynamic wetting and hence limiting the scope for contact line pinning. The experimental results achieved by the deposition with the SoS technology of PEDOT:PSS, PEIE and a ZnO-PEIE composite exhibit a thickness of the deposited films which could be easily controlled by the varying solution concentration, resulting in a minimum thickness of 15.8 nm, 4.8 nm and 15.3 nm, respectively. This shows the versatility of the Spray-on-Screen coating to deposit ultra-low thin coatings from

various materials. This is achieved irrespective of the kind of solvent and the substrate size and type (ITO or SY) being used. Especially deposition of ultra-thin films over the soft substrate like SY makes it possible to deposit on flexible substrates (for flexible electronics applications), but proper pre-treatment of the substrate should be done (UV ozone etc). The thickness could be tailor-made over a wide area of depositions with very smooth layers (1.1 nm, 0.3 nm and 2.6 nm, respectively, for PEDOT:PSS, PEIE and ZnO-PEIE composite films on super yellow substrates) opening up a new fabrication method for a wide range of application areas. By combining the microdroplet generation from USSC and the dynamic wetting by applying an external force on a screen mesh, ultra-thin coatings over a large area could be achieved. This nature-inspired, technology outperforms spin coating on the size of the substrate, outperforms other solution processing technologies for the thickness and roughness of the applied coating and outperforms vacuum deposition in view of the ease and cost of the process. Therefore, SoS is a promising technology for the deposition of ultra-thin coatings (<15 nm) on large areas, especially on heat-sensitive substrates for a variety of applications. The suitability of the SoS to build thin films for a potential application is shown by building small area OLEDs with both PEDOT:PSS and ZnO-PEIE nanocomposite with SoS. The devices outperformed the evaporated devices and equal the spin coated devices. This shows the possibility of building highly efficient devices with SoS and also the large area devices built by SoS exhibit uniform illumination with the same J-V behaviour as that of small area devices showing the uniformity in the thickness which controls the charge injection. Therefore, among many, organic electronics (such as OLEDs, tandem solar cells) and battery research are the frontrunners in adopting the SoS in the near future.

## Methods

**Materials**. ITO coated glass substrates were purchased from BIOTAIN Hongkong co. limited (thickness of 135 ± 5 nm, with a resistance of 10 ~ 15 $\Omega q^{-1}$ with a transmittance of >85%). poly(3,4-ethylenedioxythiophene) polystyrene sulfonate (PEDOT:PSS, grade Al 4083) were purchased from Heraeus. Super yellow light emitting PPV copolymer (SY), Polyethylenimine, 80% ethoxylated solution (PEIE, 37% wt % in $H_2O$), Zinc oxide nanoparticle ink with a particle diameter of 10–15 nm dispersed in IPA were purchased from Aldrich.

All the samples are cleaned according to the following protocol: ultrasonication-30 min in soap water, 20 min in demineralized water, 10 min in acetone and 10 min in isopropanal alcohol. Then the samples are dried with nitrogen gas. Finally, all the samples are UV-Ozone treated for 30 min.

**Thin film fabrication**. An ultrasonic spray coater from Sono-Tek corporation, equipped with an impact nozzle was used to spray coat. A programmable 25 ml syringe pump as a reservoir for the materials of interest, which is connected to the atomizing nozzle by tubing. The USSC nozzle was actuated at 120 kHz frequency at a generator power of 3 W. For all the set of experiments, a path speed of 10 mm s$^{-1}$, nitrogen shroud pressure of 1.5 kPa above the atmosphere is maintained. The polyester screen from Mediascreen is first cleaned by water then followed by IPA. Then the mesh is dried by using nitrogen gas for 2 min.

**Characterization**. The surface morphology and the thickness of the thin films were measured using a JPK-Bruker Atomic force microscope (AFM) with atomic resolution less than 0.030 nm and ultra-low noise level of cantilever deflection detection system which is less than 2 pm RMS; silicon ACTA tips on AC mode in the air with cantilever length ~125 μm, spring constant ~40 Nm$^{-1}$, and resonance frequency ~300 kHz were used. Force Spectroscopy was performed by using an Au-coated tip (Supplementary Fig. 18) which was calibrated using a clean ultra-flat gold substrate (Supplementary Fig. 19). On each region of 10 μm$^2$, a grid of 100 points was set and force curves were taken at each of those points applying 2 nN. Young's modulus was calculated using the data processing software of the AFM applying the Hertz/Snedon model. All measurements were carried out in contact mode in air. Scanning electron microscopy (SEM) images were recorded by using Zeiss 450 gemini 2 FEG-SEM. Contact angle measurements were conducted by using Data physics OCA 15 Plus and the results were visualized and analysed using SCA 20 software. Zeiss Axiovert 40 MAT, inverted microscopy along with Zeiss Axiocam 305 colour are utilised for the optical microscopy images.

## Data availability

Data sets generated during the current study are available from the corresponding author on reasonable request.

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

## Acknowledgements

The authors would like to thank the financial contribution from BOF UHasselt in project number BOF18NI05. Authors also would like to thank Prof. Dr. Werner Steffen, Max Planck institute for polymer research, for his insights on manuscript preparation.

## Author contributions

R.S.N. and W.D. conceptualized, designed and performed this research. W.D. supervised the work. R.S.N. has performed AFM, thickness measurements, contact angle measurements, and surface energy measurements. P.V. has measured the size of the microdroplets by capturing the microdroplets' videos and images. A.V.R. has conducted Young's modulus measurements under the supervision of F. R. P.N. has drawn and rendered the graphical abstract of the SoS technology. J.D. performed SEM images on the deposited samples. R.S.N. has drafted the manuscript under the guidance of F.R. and W.D. All the authors discussed the results and commented on the manuscript.

## Competing interests

R.S.N. and W.D., UHasselt and IMEC have a patent pending on the Spray-on-Screen technology for fabricating the ultra-thin films. The European patent application number is EP 21209014.6 and the United States patent number is US 17988874. The other authors declare no competing interests.
