## [Peer Review File · Communications Engineering]

Deposition of ultra-thin coatings by a nature-inspired Spray-on-Screen technologyReviewers' comments:

Reviewer #1 (Remarks to the Author):

This manuscript is an experimental and partially theoretical study on depositing nanoscale thick coatings on typical thin film electronic substrates using a combination of screen-printing and ultrasonic droplet spray technology. The combination results in a dynamic wetting pattern that can be used to coat surfaces with conductive polymers or with nanoparticle suspension in water or alcohol carriers. The authors used AFM extensively to characterize their coatings but also demonstrated applications such as OLED and compared their performance with spin coated or vapor deposited counterparts. In general, this reviewer had no major technical or scientific issues with the manuscript and is in the opinion that revisions are definitely needed to improve the quality of the work and make it eventually suitable for publication. The authors should address the following questions, comments and concerns in a revised version as follows:

1. Can the authors provide SEM images with EDX (EDS) elemental signals tracing Sulfur, Zinc and possibly nitrogen from the coated substrates and discuss the observations?
2. It would be great if for the ZnO coatings the EDS elemental maps are compared between their coating technology and spin coating for instance?
3. Can these coatings technology be applied to flexible substrates? Please comment and discuss.
4. What is not clear is that if the coatings are conformal then you may be measuring the roughness of the substrates? Please distinguish the roughness of the ITO surface from the deposited coatings.

Reviewer #2 (Remarks to the Author):

The authors in this manuscript reports on a novel methods to achieve homogeneous ultra thin films on different type of substrates. The methodology here proposed is based upon a modified ultrasonic spray coating and it is suitable for large area applications. The findings here reported appears to be original and potentially very interesting for the scientific community. Characterization of the deposited coatings was carried out by SEM and AFM investigations providing the homogeneous morphology of the coatings as well as a reduced surface roughness.

The method employed is sufficiently described on both theoretical and practical aspects and the production mechanism appears to be reasonable and in agreement with the experimental evidences reported.

On the bases of these considerations it is my opinion that the paper can be positively considered for publication.

At the outset, the authors are grateful for the comments made by the reviewers and we highly appreciate their suggestions and comments. We have performed experiments to answer the queries by the reviewer and the answers are given below.

1. Can the authors provide SEM images with EDX (EDS) elemental signals tracing Sulphur, Zinc and possibly nitrogen from the coated substrates and discuss the observations?

The comment by the reviewer is a good suggestion to show the readers that indeed we have these elements which are the signature of deposited material over any given substrate.

Label A: ZnO + PEIE on ITO coated glass substrate

The above EDX scan is performed at 5.0 kV. For the EDX measurements ITO coated glass substrates were used, over this substrate, PEIE-ZnO nanocomposite layers were coated by spray on screen. PEIE has nitrogen. The EDX results show the presence of nitrogen and zinc. EDX also show the presence of Indium and silicon, it is because of the base substrate.

Label A: PEDOT PSS on ITO coated glass substrate

The above EDX scan is performed at 10 kV. For the EDX measurements, ITO coated glass substrates were used, over this substrate, PEDOT:PSS layers were coated by spray on screen. The EDX shows the presence of sulphur which is a dopant in the organic semiconductor. Since the scan was performed at 10 kV, the contribution of components which makes the composition of ITO-coated glass substrate is also visible. The borosilicate glass

has the contribution from sodium and aluminium along with boron and silicon. These obtained EDX results are added to the supplementary information in Figures S 18 and 19 and also updated in the main article.

2. It would be great if for the ZnO coatings the EDS elemental maps are compared between their coating technology and spin coating for instance?

The precursor solution used for the spray on screen technology and the spin coating is the same, the EDX performed over the spray on screen does not show any impurities and hence the sample deposited with the spin coating will not be different.

3. Can these coatings technology be applied to flexible substrates? Please comment and discuss.

It is a great comment by the reviewer: the flexible substrates being used in the research community involves normal cleaning procedures but includes additional surface treatment like oxygen plasma treatment, UV-Ozone treatment, surface roughening or additional adhesion promoters. Our study has demonstrated successful deposition of the PEIE-ZnO nanocomposite onto low surface energy substrates, such as super yellow, using the spray-on-screen technique. However, for super hydrophobic flexible substrates like polyethylene terephthalate, it is advisable to perform pre-treatment similar to other coating technologies to ensure uniform deposition of layers when using the spray-on-screen method.

4. What is not clear is that if the coatings are conformal then you may be measuring the roughness of the substrates? Please distinguish the roughness of the ITO surface from the deposited coatings.

Based on the great suggestion by the reviewer, we have deposited the PEDOT:PSS and PEIE-ZnO nanocomposite over the ITO coated glass substrate and measured the surface roughness with the Bruker AFM. The results are shown in the figure as well as in the table below. The results clearly indicate the distinction between all the substrates, the ITO being rough and upon depositing PEDOT:PSS and PEIE-ZnO nanocomposite the RMS roughness reduces. These results show that indeed we have the layer deposited on the ITO substrate.

Surface roughness comparison of the spray on screen deposited PEDOT:PSS and PEIE-ZnO nanocomposite in comparison with the bare ITO substrate.

ITO	PEDOT:PSS on ITO	PEIE-ZnO NC on ITO
3.210 nm	1.00 nm	2.69 nm

These results are added to the supplementary information in Figure S20 and also updated in the main article.

REVIEWERS' COMMENTS:

Reviewer #2 (Remarks to the Author):

The revisions and the response to the examiner comments appear to be satisfactory. The revised version may be acceptable for publication without further amendments.